# Predictors of post-pericardiotomy syndrome after native valve-sparing aortic valve surgery

**Theresa Holst**◉*, **Lisa Müller, Noureldin Abdelmoteleb, Sina Stock, Tatiana M. Sequeira Gross, Evaldas Girdauskas**◉*

Department of Cardiothoracic Surgery, Augsburg University Hospital, Augsburg, Germany

* theresa.holst@uk-augsburg.de (TH); evaldas.girdauskas@uk-augsburg.de (EG)

## Abstract

### Background

We aimed to determine the rate and impact of post-pericardiotomy syndrome after native valve-sparing aortic valve surgery and the perioperative factors associated with its occurrence.

### Methods

All consecutive patients who underwent native valve-sparing aortic valve surgery (i.e., repair ± ascending aorta replacement, valve-sparing root replacement, Ross procedure ± ascending aorta replacement) at our institution between January 2021 and August 2023 served as our study population. Post-pericardiotomy syndrome was diagnosed if patients showed at least two of the following diagnostic criteria: evidence of (I) new/worsening pericardial effusion, or (II) new/worsening pleural effusions, (III) pleuritic chest pain, (IV) fever or (V) elevated inflammatory markers without alternative causes. A logistic regression model was calculated.

### Results

During the study period, 91 patients underwent native valve-sparing aortic valve surgery. A total of 21 patients (23%) developed post-pericardiotomy syndrome early after surgery (PPS group). The remaining 70 patients (77%) showed no signs of post-pericardiotomy syndrome (non-PPS group). Multivariate logistic regression revealed blood type O (OR: 3.15, 95% CI: 1.06–9.41, p = 0.040), valve-sparing root replacement (OR: 3.12, 95% CI: 1.01–9.59, p = 0.048) and peak C-reactive protein >15 mg/dl within 48 hours postoperatively (OR: 4.27, 95% CI: 1.05–17.29, p = 0.042) as independent risk factors. 73% (8/11) of patients displaying all three risk factors, 60% (9/15) of patients with blood type O and valve-sparing root replacement, 52% (11/21) of patients with blood type O and early postoperative peak C-reactive protein >15 mg/dl and 45% (13/29) of patients with early postoperative peak C-reactive protein >15 mg/dl and valve-sparing root replacement developed post-pericardiotomy syndrome.

**Data Availability Statement:** Data cannot be shared publicly for proprietary reasons. Data are available from the Ethics Committee of the Ludwig Maximilian University of Munich (contact via

ethikkommission@med.uni-muenchen.de) for researchers who meet the criteria for access to confidential data.

**Funding:** The author(s) received no specific funding for this work.

**Competing interests:** The authors have declared that no competing interests exist.

## Conclusion

In summary, blood type O, valve-sparing root replacement and peak C-reactive protein >15 mg/dl within 48 hours postoperatively are significantly associated with post-pericardiotomy syndrome after native valve-sparing aortic valve surgery. Particularly, the presence of all three risk factors is linked to a particularly high risk of post-pericardiotomy syndrome.

## Introduction

Post-pericardiotomy syndrome (PPS) is an autoinflammatory/autoimmune syndrome with pleuro-pericardial involvement affecting approximately 10–30% of patients in the early postoperative period (i.e. up to three months) after cardiac surgery [1, 2]. It appears to be initiated by a combination of perioperative bleeding in the pericardial and pleural cavities and surgical damage to the pericardium, myocardium and/or pleurae, presumably triggering the release of antigens, the production of autoantibodies and a subsequent local and systemic inflammatory response. Yet, the exact etiopathogenesis is still not well understood [3]. Common clinical features of PPS include fever and elevated C-reactive protein (CRP) levels without alternative causes, pericardial or pleuritic chest pain, new onset of atrial fibrillation as well as evidence of pericardial and pleural effusions, but clinical presentation is variable and severity ranges from only mild symptoms to severe pleuro-pericardial involvement requiring reintervention (i.e., pleurocentesis or thoracoscopy, pericardiocentesis or pericardiotomy) [3]. Basic diagnostic evaluation, therefore, includes physical examination, routine blood work, electrocardiogram, transthoracic echocardiography (TTE) including lung ultrasound and chest X-ray according to current European guidelines [4]. Cardiac computed tomography (CT) and magnetic resonance imaging may be added to evaluate extension and hemodynamic impact of effusions and to guide intervention [3]. Pharmacological treatment strategies comprise nonsteroidal anti-inflammatory drugs, more specifically indomethacin or ibuprofen, in combination with colchicine [4–6]. The addition of corticosteroids may be considered, but evidence is scarce and there are concerns about higher recurrence rates [7, 8]. The general prognosis of PPS is good as it resolves in approximately 95% of patients without any sequelae or residues. However, its major complications (e.g. cardiac tamponade, respiratory impairment and constrictive pericarditis) are associated with increased morbidity and mortality [7]. Moreover, recurrences are observed in a few cases [1]. To date, factors associated with the occurrence of PPS have not been well investigated and results of published studies are somewhat conflicting [3]. It appears that the incidence is higher in younger patients [9]. Moreover, PPS seems more common after aortic and valvular procedures compared to coronary artery bypass grafting [7, 9]. Consequently, patients selected for native valve-sparing aortic valve (AV) surgery (i.e. isolated AV repair, AV repair and ascending aorta replacement, valve-sparing root replacement, isolated Ross procedure or Ross procedure and ascending aorta replacement) that are generally < 65 years old could be at particularly high risk of developing PPS. To the best of our knowledge, no comprehensive study has systematically investigated the prevalence of PPS among this specific patient population. The aim of this study, therefore, was to determine the prevalence and impact of PPS after native valve-sparing AV surgery. Moreover, we aimed to evaluate the perioperative factors associated with its occurrence, thereby identifying patients that might benefit from preventive medication.

## Materials and methods

### Study population

From January 2021 to August 2023, a total of 91 non-elderly patients (mean age: 46±12 years, 89% male) were primarily referred to our institution for native valve-sparing surgery for severe AV disease (i.e., isolated aortic regurgitation, isolated or predominant bicuspid or unicuspid aortic stenosis). In patients presenting with isolated aortic regurgitation and sufficient AV tissue quality either AV repair (n = 23) or valve-sparing root replacement (i.e., reimplantation according to David using a sinus Dacron graft, n = 35) was performed depending on the aortic root diameter. The Ross procedure (n = 7) was opted for patients presenting with isolated or predominant unicuspid or bicuspid aortic stenosis and well-functioning pulmonary valves. After harvesting of the pulmonary autograft, the right ventricular outflow tract was reconstructed using a cryopreserved or decellularized pulmonary homograft. In case of simultaneous ascending aorta aneurysm, AV repair (n = 13) and the Ross procedure (n = 13) were accompanied by concomitant replacement of the ascending aorta using a straight Dacron graft. Preoperative diagnostics revealed concomitant pathologies requiring treatment (i.e., at least moderate mitral or tricuspid valve disease, $\geq$ 70% coronary artery stenosis, atrial septal defect, patent foramen ovale, persistent or paroxysmal atrial fibrillation) in 18 patients. Concomitant procedures, hence, included mitral and/or tricuspid valve surgery (n = 10), left atrial appendage occlusion (n = 7), coronary artery bypass grafting (n = 5), closure of an atrial septal defect or a patent foramen ovale (n = 2) and left atrial ablation for atrial fibrillation (n = 2). Pre-, peri- and postoperative data from all patients' hospital charts was retrospectively gathered in an anonymized fashion and reviewed for the current study in October 2023 after approval by the ethics committee of the Ludwig Maximilian University of Munich, Germany (23–0323). Written informed consent was obtained from all subjects prior to inclusion.

### Diagnosis and management of post-pericardiotomy syndrome

PPS was diagnosed if patients showed at least two of the following diagnostic criteria as adapted from Imazio et al. [10]: (I) evidence of new or worsening pericardial effusion on TTE, (II) evidence of new or worsening pleural effusions on lung ultrasound and chest X-ray, (III) pleuritic chest pain, (IV) fever or (V) elevated inflammatory markers without alternative causes. Moreover, nosocomial systemic infection had to be ruled out. Once the diagnosis was confirmed, anti-inflammatory medical treatment using ibuprofen (600 mg 3 times per day) in combination with colchicine (0.5 mg twice a day) was administered for a minimum of 14 days. Rarely, low-dose corticosteroids (0.25–0.5 mg/kg/d) were added as well. In case of clinical deterioration, such as progressive respiratory distress with increasing need for oxygen administration or clinical suspicion of cardiac tamponade, a CT scan was performed to precisely evaluate extension of pleural and pericardial effusions and to guide urgent relief via thoracocentesis in case of isolated large pleural effusions, pericardiocentesis or pericardiotomy in case of isolated pericardial effusion or video-assisted thoracoscopy for combined pleuro-pericardial involvement.

### Statistical analysis

Categorical variables are expressed as absolute and relative frequencies and comparisons were made using Fisher's exact test and adopting a two-tailed significance for the $p$-value. Normally distributed continuous variables are presented as mean ± standard deviation. Between-group comparisons were performed using an unpaired two-sample $t$-test. Continuous variables (e.g. CRP level) were converted into binary variables using cut-off values based on clinical

significance, if deemed necessary for further analysis. Potential pre-, peri- and postoperative risk factors (i.e., candidates) identified by significant differences in the univariate analysis were further tested using a multivariate binary logistic regression model to explore independent variables associated with the occurrence of PPS. Variables in the logistic regression model were selected by block entry and odds ratio (OR), 95% confidence intervals (CIs) and p-values were reported. All p-values <0.05 were considered statistically significant. IBM SPSS Version 29.0.1.0 software (IBM Corp., Armonk, New York, USA) was used for all statistical analyses.

## Results

### Prevalence of post-pericardiotomy syndrome in the study cohort

A total of 21 patients (23%) developed PPS during the early postoperative course and underwent anti-inflammatory pharmacological treatment (PPS group). The remaining 70 patients (77%) remained free from any signs/features of PPS (non-PPS group). PPS was diagnosed during the initial hospital stay in almost all PPS patients. Nevertheless, 2 patients experienced an uneventful initial postoperative stay, but were re-admitted at 2 and 3 weeks postoperatively for signs of cardiac tamponade. Pericardial involvement was frequent and observed in 16 PPS patients (76%). Intervention/surgery for isolated worsening partly or entirely serous pericardial effusion was performed in 4 cases. Pleural involvement was seen in 14 PPS patients (67%) and drainage of isolated large partly or entirely serous pleural effusions was necessary in 5 cases. Two patients required combined evacuation of partly or entirely serous pericardial and pleural effusions. Moreover, 9 PPS patients (43%) had new onset of atrial fibrillation and received antiarrhythmic pharmacological treatment (in contrast to only 16 non-PPS patients (23%)).

### Pre- and perioperative patient characteristics

Pre- and perioperative characteristics of the PPS and the non-PPS patients are displayed in Table 1. No significant differences in age, sex distribution, comorbidities, cardiac risk factors and perioperative risk scores could be detected between both groups. Not a single patient had a history of pericarditis or prednisolone treatment. Furthermore, cardiopulmonary bypass and cross-clamp times did not differ between both groups and the relative frequency of simultaneous surgical procedures was similar among PPS and non-PPS patients. However, the relative frequency of blood type O (57% vs. 30%, p = 0.037), mini-sternotomy (81% vs. 51%, p = 0.022) and valve-sparing root replacement (67% vs. 30%, p = 0.004) was significantly higher and the relative frequency of tricuspid AV morphology (48% vs. 24%, p = 0.056) also remarkably higher in PPS vs. non-PPS patients.

### Postoperative laboratory values

During the postoperative course, the peak white blood cell count, highest temperature, peak creatinine and peak creatine kinase within the first 48 hours postoperatively, fluid balance during stay on intensive/intermediate care unit, duration of the first postoperative rise of the white blood cells and CRP as well as the absolute max white blood cell count at the end of the first rise after surgery were similar in both groups. However, peak CRP within the first 48 hours postoperatively (i.e., the highest recorded CRP value within the first 48 hours after surgery; 22.3±7.2 vs. 16.6±6.6 mg/dl, p = 0.001) and absolute max CRP value (i.e., the highest recorded CRP value at the end of the first continuous rise after surgery; 25.0±7.9 vs. 18.3±7.3 mg/dl, p<0.001) were significantly higher in PPS vs. non-PPS patients (see Table 2). In fact,

**Table 1. Pre- and perioperative patient characteristics.**

| | All (n = 91) | No PPS (n = 70) | PPS (n = 21) | p-value |
|---|---|---|---|---|
| Age (years) | 46±12 | 46±12 | 46±13 | 0.855 |
| Female sex | 10 (11%) | 7 (10%) | 3 (14%) | 0.692 |
| Comorbidities/risk factors | | | | |
| BMI (kg/m$^2$) | 26±4 | 26±4 | 26±3 | 0.296 |
| Creatinine (mg/dl) | 0.95±0.29 | 0.96±0.31 | 0.89±0.18 | 0.357 |
| proBNP (ng/l) | 509±1201 | 566±1333 | 321±569 | 0.417 |
| Hypertension | 46 (51%) | 36 (52%) | 10 (48%) | 0.807 |
| Diabetes | 3 (3%) | 3 (4%) | 0 (0%) | 1.000 |
| Hyperlipidemia | 17 (19%) | 15 (21%) | 2 (10%) | 0.341 |
| CAD | 7 (8%) | 7 (10%) | 0 (0%) | 0.195 |
| s/p stroke | 6 (7%) | 4 (6%) | 2 (10%) | 0.619 |
| s/p pericarditis | 0 (0%) | 0 (0%) | 0 (0%) | 1.000 |
| s/p cardiac surgery | 2 (2%) | 2 (3%) | 0 (0%) | 1.000 |
| Known allergies | 25 (28%) | 20 (29%) | 5 (24%) | 0.785 |
| Prednisolone treatment | 0 (0%) | 0 (0%) | 0 (0%) | 1.000 |
| STS-PROM (%) | 1.06±1.01 | 1.11±1.05 | 0.87±0.85 | 0.342 |
| EuroSCORE II (%) | 1.88±2.18 | 2.01±2.43 | 1.42±0.87 | 0.280 |
| Type of blood | | | | |
| O | 33 (36%) | 21 (30%) | 12 (57%) | **0.037** |
| AB | 3 (3%) | 3 (4%) | 0 (0%) | 1.000 |
| A | 41 (45%) | 34 (49%) | 7 (33%) | 0.318 |
| B | 14 (15%) | 12 (17%) | 2 (10%) | 0.508 |
| AV morphology | | | | |
| Uni-/bicuspid | 64 (70%) | 53 (75%) | 11 (52%) | **0.056** |
| Tricuspid | 27 (30%) | 17 (24%) | 10 (48%) | **0.056** |
| Surgical access | | | | |
| Mini-sternotomy | 53 (58%) | 36 (51%) | 17 (81%) | **0.022** |
| Median sternotomy | 38 (42%) | 34 (49%) | 4 (19%) | **0.022** |
| Type of surgery | | | | |
| Isolated AV repair | 23 (25%) | 20 (29%) | 3 (14%) | 0.257 |
| VSRR | 35 (39%) | 21 (30%) | 14 (67%) | **0.004** |
| AV repair + AA replacement | 13 (14%) | 11 (16%) | 2 (10%) | 0.725 |
| Isolated Ross | 7 (8%) | 6 (9%) | 1 (5%) | 1.000 |
| Ross + AA replacement | 13 (14%) | 12 (17%) | 1 (5%) | 0.285 |
| Use of Dacron material | 61 (67%) | 44 (63%) | 17 (81%) | 0.185 |
| Simultaneous surgical procedures | 18 (20%) | 15 (21%) | 3 (14%) | 0.551 |
| CPB duration (min) | 140±45 | 145±48 | 127±32 | 0.052 |
| ACC duration (min) | 95±33 | 98±34 | 86±26 | 0.158 |

Data presented as mean±SD or absolute (relative) frequencies. Data derived from unpaired 2-sample t-test or Fisher's exact test. AA: ascending aorta; ACC: aortic cross-clamp; AV: aortic valve; BMI: body mass index; BNP: brain natriuretic peptide; CAD: coronary artery disease; CPB: cardiopulmonary bypass; PPS: post-pericardiotomy syndrome; PROM: predicted risk of mortality; s/p: status post; VSRR: valve-sparing root replacement.

**Table 2. Postoperative laboratory values.**

|  | All (n = 91) | No PPS (n = 70) | PPS (n = 21) | p-value |
|---|---|---|---|---|
| Absolute max postop CRP (mg/dl) | 19.6±7.9 | 18.3±7.3 | 25.0±7.9 | <**0.001** |
| Duration of first postop rise in CRP (days) | 2.6±0.8 | 2.5±0.7 | 2.9±0.9 | 0.088 |
| Absolute max postop white blood cell count (/nl) | 16.3±4.6 | 16.3±4.8 | 16.4±3.9 | 0.927 |
| Duration of first postop rise in white blood cells (days) | 0.7±0.9 | 0.7±0.9 | 1.0±1.0 | 0.264 |
| Peak CRP within the first 48 hours postop (mg/dl) | 17.9±7.1 | 16.6±6.6 | 22.3±7.2 | **0.001** |
| Peak white blood cell count within the first 48 hours postop (/nl) | 16.1±5.2 | 16.0±5.5 | 16.3±3.9 | 0.820 |
| Peak creatinine within the first 48 hours postop (mg/dl) | 1.09±0.38 | 1.08±0.42 | 1.09±0.25 | 0.962 |
| Peak CK within the first 48 hours postop (U/l) | 1172±1553 | 1250±1747 | 911±479 | 0.384 |
| Fluid balance on ICU/IMC (ml) | 1453±7150 | 685±3100 | 4119±13936 | 0.300 |
| Need for blood transfusions | 22 (24%) | 16 (23%) | 6 (29%) | 0.574 |
| Length of hospital stay | 8.2±4.8 | 8.0±5.2 | 9.1±3.0 | 0.363 |

Data presented as mean±SD or absolute (relative) frequencies. Data derived from unpaired 2-sample t-test or Fisher's exact test. CK: creatine kinase; CRP: C-reactive protein; ICU: intensive care unit; IMC: intermediate care unit; PPS: post-pericardiotomy syndrome.

34% of patients with a peak CRP > 15 mg/dl within the first 48 hours postoperatively and 50% of patients with an absolute max postoperative CRP > 25 mg/dl developed PPS.

## Predictors of post-pericardiotomy syndrome

Univariate binary logistic regression identified the following variables as significantly associated with PPS: blood type O (OR: 3.11, 95% CI: 1.14–8.49, p = 0.027), tricuspid AV morphology (OR: 2.83, 95% CI: 1.03–7.83, p = 0.044), mini-sternotomy (OR: 4.01, 95% CI: 1.23–13.14, p = 0.022), valve-sparing root replacement (OR: 4.67, 95% CI: 1.65–13.22, p = 0.004), peak CRP > 15 mg/dl within the first 48 hours postoperatively (OR: 6.00, 95% CI: 1.62–22.21, p = 0.007) as well as absolute max postoperative CRP > 25 mg/dl (OR: 7.13, 95% CI: 2.45–20.71, p<0.001). All variables with a p-value less than 0.10 were entered in the multivariate analysis. Tricuspid AV morphology and mini-sternotomy were excluded because of their significant association with valve-sparing root replacement (contingency coefficient: 0.465, p<0.001 and contingency coefficient: 0.207, p = 0.004, respectively) and absolute max postoperative CRP > 25 mg/dl was excluded because of its significant association with peak CRP > 15 mg/dl within the first 48 hours postoperatively (contingency coefficient: 0.472, p<0.001). In the multivariate binary logistic regression, blood type O (OR: 3.15, 95% CI: 1.06–9.41, p = 0.040), valve-sparing root replacement (OR: 3.12, 95% CI: 1.01–9.59, p = 0.048) and peak CRP > 15 mg/dl within the first 48 hours postoperatively (OR: 4.27, 95% CI: 1.05–17.29, p = 0.042) were identified as independent risk factors associated with PPS after native valve-sparing AV surgery. Results are illustrated in Table 3. 73% (8/11) of patients displaying all three risk factors, 60% (9/15) of patients with blood type O and valve-sparing root replacement, 52% (11/21) of patients with blood type O and early postoperative peak CRP > 15 mg/dl and 45% (13/29) of patients with early postoperative peak CRP > 15 mg/dl and valve-sparing root replacement developed PPS (see Fig 1).

## Discussion

PPS is a common complication after cardiac surgery in children and adult patients with an incidence of almost 30% in previous studies [1, 2, 11–15]. Most PPS patients require intensified medical attention, specific medical and interventional treatment, hospital stay

**Table 3. Risk factors for post-pericardiotomy syndrome after native valve-sparing aortic valve surgery.**

| Risk factor | Univariate analysis | | | Multivariate analysis | | |
|---|---|---|---|---|---|---|
| | OR | 95% CI | p-value | OR | 95% CI | p-value |
| Blood type O (vs. other types of blood) | 3.11 | 1.14-8.49 | 0.027 | 3.15 | 1.06-9.41 | 0.040 |
| Tricuspid AV (vs. unicuspid/bicuspid) | 2.83 | 1.03-7.83 | 0.044 | | | |
| Mini-sternotomy (vs. median sternotomy) | 4.01 | 1.23-13.14 | 0.022 | | | |
| VSRR (vs. other types of surgery) | 4.67 | 1.65-13.22 | 0.004 | 3.12 | 1.01-9.59 | 0.048 |
| Peak CRP >15 mg/dl within the first 48 hours postop | 6.00 | 1.62-22.21 | 0.007 | 4.27 | 1.05-17.29 | 0.042 |
| Absolute max postop CRP >25 mg/dl | 7.13 | 2.45-20.71 | <0.001 | | | |

AV: aortic valve; CI: confidence interval; CRP: C-reactive protein; OR: odds ratio; VSRR: valve-sparing root replacement.

prolongation or readmission during follow-up. All these factors have a negative impact on the patients' recovery process and may hinder enhanced recovery after surgery (ERAS) protocols. Similar to previous reports, nearly 1 in 4 patients undergoing native valve-preserving AV surgery developed PPS requiring additional medical/surgical treatment in our series. Intervention or surgery for worsening pericardial and pleural effusions was performed in 11 patients (52%) (i.e., severe PPS): 4 patients needed pericardiocentesis or pericardiotomy for isolated hemodynamically relevant partly or entirely serous pericardial effusion, 5 patients required drainage of isolated large partly or entirely serous pleural effusions and 2 patients needed combined evacuation of partly or entirely serous pericardial and pleural effusions via video-assisted thoracoscopy. 12% of patients in our total study population had severe PPS which is markedly higher than in a previous study reporting on isolated AV replacement surgery [13]. These divergent findings may be potentially explained by distinct patients' characteristics as well as different

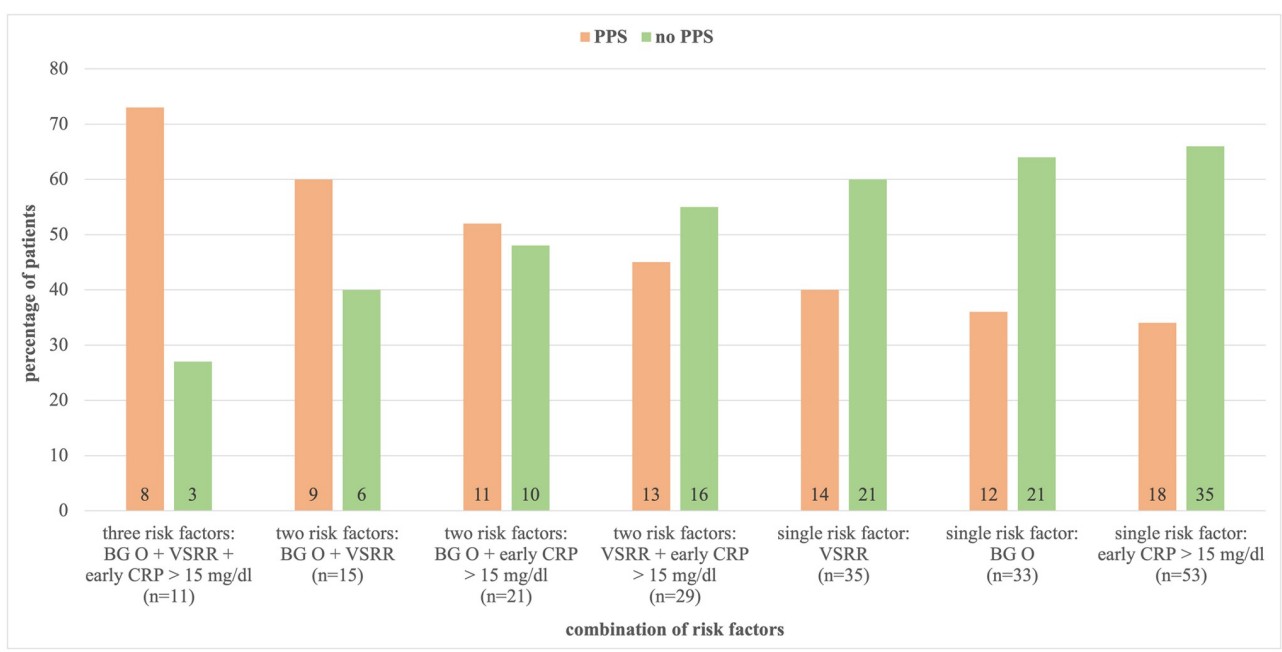

**Fig 1. Absolute and relative frequency of patients developing post-pericardiotomy syndrome (PPS) stratified by combination of risk factors present (i.e., all three, any combination of two, any one).** BG: blood group/type of blood; CRP: C-reactive protein; VSRR: valve-sparing root replacement.

surgical maneuvers in both studies. Our patients were remarkably younger, more frequently male and had more often simultaneous aortic root surgery. Yet, the high rate of severe PPS after native valve-sparing AV surgery clearly underlines the need for a risk factor analysis to determine patients that might especially benefit from early preventive measures.

Currently, specific risk factors associated with the occurrence of PPS are insufficiently defined due to limited data and conflicting evidence from previous trials. The only consented risk factor, most authors agreed upon, is a younger age most likely due to an age-related increasing immune senescence [3, 7, 16]. Some studies reported female patients to be at higher risk for PPS which is potentially due to higher susceptibility to autoimmune diseases [3, 7, 17]. Surprisingly, we did not observe any significant impact of age and sex on the occurrence of PPS in our study. Patients of both groups were on average 46 years old and the ratio of males to females was comparable. A potential explanation might be a very specific, quite homogenous study cohort as compared to most previous studies. This explanation is in line with previous findings by van Osch and colleagues and Heching and colleagues [11, 18]. Both above-mentioned studies found no impact of age or sex on the occurrence of PPS after valvular surgery or surgical closure of an atrial septal defect [11, 18]. Other possible risk factors, that have been mentioned and controversially discussed in the literature, include lower body weight, a history of pericarditis, past or present prednisolone treatment, reduced preoperative renal function and an increased postoperative transfusion requirement [5, 12]. Diabetes, on the other hand, has been described to have a protective effect on the occurrence of PPS [12]. We did not observe significant differences in any of the above-mentioned variables between patients with PPS vs. those without PPS.

There is evidence that patients with certain blood types may be predisposed to infections and autoimmune conditions. Yet, the results of studies examining the link between ABO blood group and autoimmunity have been inconsistent [19–22]. To our knowledge, only two previous studies reported a potential association of ABO blood group with PPS: Maisch et al. found anti-endothelial antibodies to be more frequent in patients who developed PPS than in those who did not. Moreover, they suggested an association of the prevalence of this specific form of autoantibodies after cardiac surgery with blood groups A, B and O, but not with AB [23]. Miller et al. reported patients with blood type B to be at higher risk of PPS compared to other blood types. Again, blood type AB had the lowest risk of PPS [5]. Consistent with previous findings, not a single PPS patient from our cohort had blood type AB. In contrast to both aforementioned studies, our results, however, indicated a predisposition of blood type O towards PPS. More comprehensive studies are needed to further characterize the link between ABO blood group and the risk of PPS.

Valve-sparing root replacement (i.e., the reimplantation procedure) was independently associated with PPS in our study. AV repair (with or without concomitant replacement of the ascending aorta) and valve-sparing root replacement are routinely performed through a mini-sternotomy (i.e., partial upper L sternotomy in the third intercostal space) at our institution. Yet, the extent of surgical trauma is greater in valve-sparing root replacement than in AV repair due to obligatory deep aortic root dissection and additional surgical steps during reimplantation. As surgical trauma may promote pleuro-pericardial antigen presentation to the immune system, triggering the production of autoantibodies and subsequent PPS [3], it seems logical that more extensive injury leads to a higher incidence of PPS after valve-sparing root replacment. The Ross procedure (with or without concomitant replacement of the ascending aorta) is naturally even more invasive due to surgical maneuvers on both the aortic and pulmonary valve. Moreover, this procedure is still performed through a median sternotomy at our institution which is associated with larger surgical trauma as compared to mini-sternotomy. Therefore, an increased risk of PPS in patients after valve-sparing root replacement compared

to the Ross procedure at first seems somewhat counterintuitive, but could possibly be explained by the distinctly different placement of chest drains in mini- vs. median sternotomy: Procedures performed through a median sternotomy, such as the Ross procedure, commonly involve the placement of two pericardial drains, one placed retrosternally in the anterior pericardium and one placed along the left ventricle in the retrocardiac space. In contrast, in minimally-invasive procedures with mini-sternotomy as surgical access, such as valve-sparing root replacement, generally only a single drain is placed in the anterior pericardium. This could result in insufficient drainage of the posterior aspect of the pericardium and contribute to retained blood syndrome which is believed to be an important trigger of the exaggerated immune response and inflammation seen in PPS [3, 24, 25]. Moreover, the risk of accidental/unnoticed pleural opening which has been described as a predisposing factor for PPS [1, 7] might occur more frequently in minimally-invasive procedures than in median sternotomy due to limited exposure and view of the surrounding structures.

The evidence on the value of classical inflammatory markers in predicting PPS after cardiac surgery is very scarce. Köhler et al. observed no significant difference in postoperative CRP level between PPS and non-PPS patients after various types of cardiac surgery [26]. On the other hand, the white blood cell count on postoperative day one was independently associated with PPS after on-pump coronary artery bypass grafting in a study by Sevuk et al. [27]. As opposite, we did not find any difference in postoperative white blood cell count in PPS vs. non-PPS patients. However, our study revealed significantly higher peak postoperative CRP levels in patients who developed PPS. Moreover, early postoperative (i.e., within the first 48 hours after surgery) CRP peak > 15 mg/dl was found to be independently associated with the occurrence of PPS. As CRP is well-recognized as a sensitive biomarker of any type of inflammation, including immune-mediated inflammatory disease [28], we interpreted the higher CRP in PPS patients as a reflection of the increased reactivity of the immune system despite the absence of disease-mediating pathogens.

## Limitations

The retrospective observational study design renders our analysis prone to confounding and bias. Moreover, 95% confidence intervals in our multivariate binary logistic regression model were relatively large with respect to some variables most likely due to the limited sample size and data variability. Generalization of findings to other centers and settings is, hence, limited and subsequent multi-center studies with larger and more homogenous patient collectives are required to confirm our findings. Due to the fact that our institution is a reference center for the treatment of AV disease in non-elderly patients, nearly all patients aged < 65 years presenting to our institution undergo native valve-sparing AV procedures. Although we would expect similar results after conventional AV replacement, the applicability of our current results to non-elderly patients undergoing AV replacement requires further subsequent studies. Also, postoperative follow-up of our cohort was limited to 3 months. Hence, although unlikely, the potential occurrence of late PPS (i.e., first diagnosis at later than 3 months postoperatively) was disregarded.

## Conclusion

In summary, blood type O, valve-sparing root replacement and CRP peak levels > 15 mg/dl within the first 48 hours postoperatively are significantly associated with PPS after native valve-sparing AV surgery. In particular, the presence of all three risk factors is linked to a particularly high risk of PPS. To prevent postoperative complications and enable successful ERAS protocols, early preventive anti-inflammatory treatment should be initiated in such patients.

## Author Contributions

**Conceptualization:** Theresa Holst, Noureldin Abdelmoteleb, Evaldas Girdauskas.

**Data curation:** Theresa Holst, Lisa Müller.

**Formal analysis:** Theresa Holst.

**Investigation:** Theresa Holst.

**Methodology:** Tatiana M. Sequeira Gross, Evaldas Girdauskas.

**Project administration:** Evaldas Girdauskas.

**Resources:** Evaldas Girdauskas.

**Software:** Theresa Holst, Evaldas Girdauskas.

**Supervision:** Sina Stock, Evaldas Girdauskas.

**Visualization:** Theresa Holst.

**Writing – original draft:** Theresa Holst.

**Writing – review & editing:** Lisa Müller, Noureldin Abdelmoteleb, Sina Stock, Tatiana M. Sequeira Gross, Evaldas Girdauskas.

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
