## [Decision Letter · Decision Letter 0]

4 Apr 2024

PONE-D-24-05122Predictors of post-pericardiotomy syndrome after native valve-sparing aortic valve surgeryPLOS ONE

Dear Dr. Holst,

Thank you for submitting your manuscript to PLOS ONE. After careful consideration, we feel that it has merit but does not fully meet PLOS ONE’s publication criteria as it currently stands. Therefore, we invite you to submit a revised version of the manuscript that addresses the points raised during the review process.

We look forward to receiving your revised manuscript.

Kind regards,

Eyüp Serhat Çalık

Academic Editor

PLOS ONE

Journal Requirements:

2. In the online submission form you indicate that your data is not available for proprietary reasons and have provided a contact point for accessing this data. Please note that your current contact point is a co-author on this manuscript. According to our Data Policy, the contact point must not be an author on the manuscript and must be an institutional contact, ideally not an individual. Please revise your data statement to a non-author institutional point of contact, such as a data access or ethics committee, and send this to us via return email. Please also include contact information for the third party organization, and please include the full citation of where the data can be found.

Additional Editor Comments:

I congratulate the authors for their successful treatment and work. I believe that your manuscript addresses an important issue in cardiac surgery. Your manuscript has been reviewed by three reviewers and their recommendations are as follows. In particular, reviewer 2 has significant concerns about the statistical analysis. You have patients who underwent different surgical procedures, please provide more detailed information about these subgroups and include your analysis and comments on their effects on PPS. I wish you success.

Reviewers' comments:

Reviewer's Responses to Questions

**Comments to the Author**

1. Is the manuscript technically sound, and do the data support the conclusions?

Reviewer #1: Yes

Reviewer #2: Partly

Reviewer #3: Yes

2. Has the statistical analysis been performed appropriately and rigorously? 

Reviewer #1: Yes

Reviewer #2: No

Reviewer #3: Yes

3. Have the authors made all data underlying the findings in their manuscript fully available?

Reviewer #1: No

Reviewer #2: No

Reviewer #3: Yes

4. Is the manuscript presented in an intelligible fashion and written in standard English?

Reviewer #1: Yes

Reviewer #2: Yes

Reviewer #3: Yes

5. Review Comments to the Author

Reviewer #1: Holst et al reported their work named "Predictors of post-pericardiotomy syndrome after native valve-sparing aortic valve surgery" and concluded "In summary, blood type 0, valve-sparing root replacement and peak C-reactive protein>15 mg/dl within 48 hours postoperatively are significantly associated with postpericardiotomy syndrome after native valve-sparing aortic valve surgery. Particularly, the combination of all three risk factors is linked to a prohibitive risk of postpericardiotomy syndrome.". I have the following comments:

- Please don't include number of patients in the methods and move that to abstract results.

- Please add a definition with reference to "postpericardiotomy syndrome".

- Table 3: please acknowledge the wide 95%CI in your multivariable model.

- Language revision is essential eg "blood type 0," should be "blood type O"

- In the conclusion, please revisit and edit this sentence "combination of all three risk factors is linked to a prohibitive risk of PPS"

- Please cite the following relevant randomized trial in your discussion (PMID: 34788640)

Reviewer #2: Comments：

Theresa Holst and colleagues reported on 91 consecutive patients who underwent native valve-sparing aortic valve surgery and the incidence of post-pericardiotomy syndrome (PPS) among them. I congratulate them on achieving good surgical outcomes and handling PPS well. The authors attempted to identify predictive factors for PPS through the clinical data of these patients. However, there are several issues with the article:

1. The study included 91 patients, which is a small sample size that highlighted considerable heterogeneity among the patients. They underwent various types of native valve-sparing aortic valve surgeries, and some patients also had concomitant procedures, such as mitral and tricuspid valve surgeries, and CABG. Therefore, the trauma experienced by these patients was inconsistent and difficult to assess. For example, did patients undergoing VSRR also have additional surgeries, leading to the occurrence of PPS?

2. In Table 1, the statistical methods used are confusing. For categorical tests, especially with small sample sizes, using Fisher's exact test and adopting a two-tailed significance for the P-value would be more appropriate. However, in practice, the P-value for AV morphology seems to have been calculated using a one-tailed significance from Fisher's exact test (P < 0.05), but when corrected for two-tailed significance (P = 0.056), which makes it less significant. Additionally, the P-value for VSRR, despite showing a significant difference, should be 0.004 instead of 0.002. Similar statistical calculations should be re-examined by the authors, as this could lead to entirely inconsistent conclusions.

3. Regard as the results, the authors identified some predictors for PPS, such as blood type O, VSRR, and peak CPR > 15 mg/dl within 48 hours. Although these results showed statistical differences, I have some reservations about these factors. Firstly, the authors did not explain in detail in the discussion why blood type O is a predictive factor; moreover, this result differs from previous studies (blood type B), and we are very curious about why different results emerged, or is it just a statistical coincidence? Secondly, the article explains that VSRR is a predictor for PPS but cannot be explained by surgical trauma, as ROSS surgery, which involves a similar level of trauma, did not show positive results. So, what are the specific mechanisms of VSRR leading to PPS? Besides, how is peak CPR defined? Is it limited to a continuous rise within 48 hours to reach a peak, or does the peak CPR during the entire hospital stay occur within 48 hours post-surgery? If it's the latter, this indicator's reference value for clinical practice is significantly reduced, as you never know if the next blood draw will be the peak, and PPS can already be detected using echocardiography before detecting the peak.

In summary, the heterogeneity of the study population was not addressed, there are errors in the statistical analysis, and the clinical relevance of the study's results is questionable.

Reviewer #3: The authors highlighted a common but sometimes forgotten issue in post-operative complication after the cardiac surgery. It is sometimes mild but sometimes leads to delayed discharge.

They focused this complication in the category of Valve-sparing procedure. Theri methodology seems solid and their conclusion is reasonable.

The result can be recommended for those under Valve spring procedure in early management for those undergoing the procedure with more than 2 risk factors.

Only one comment.

Please describe the RIght ventricualr outflow tract reconstruction procedure in ROss procedure.

6. PLOS authors have the option to publish the peer review history of their article (what does this mean?). If published, this will include your full peer review and any attached files.

Reviewer #1: **Yes: **Mohamed Rahouma

Reviewer #2: No

Reviewer #3: No

---

## [Author Response · Author response to Decision Letter 0]

10 Apr 2024

Editor:

Comment: I congratulate the authors for their successful treatment and work. I believe that your manuscript addresses an important issue in cardiac surgery. Your manuscript has been reviewed by three reviewers and their recommendations are as follows. In particular, reviewer 2 has significant concerns about the statistical analysis. You have patients who underwent different surgical procedures, please provide more detailed information about these subgroups and include your analysis and comments on their effects on PPS. I wish you success.

Response:

Dear Mr. Çalık,

Thank you very much for giving us the opportunity to submit a revised draft of our manuscript entitled “Predictors of post-pericardiotomy syndrome after native valve-sparing aortic valve surgery” to PLOS ONE for consideration as Original Article. We appreciate the time and effort you and the three reviewers dedicated to providing valuable feedback on our manuscript and are grateful for all the insightful comments and proposals made in order to improve our study. We have incorporated the suggestions made by the reviewers and highlighted all changes within the revised manuscript and tables files. We also responded point-by-point to all of the reviewers’ comments. Note that all page and paragraph numbers refer to the revised manuscript file with tracked changes. We hope that we could adequately deal with any of the issues raised and by doing so, could significantly improve our manuscript.

We thank you very much again for your consideration.

Yours sincerely,

Theresa Holst, on behalf of all authors

Reviewer 1:

Comment: Holst et al reported their work named "Predictors of post-pericardiotomy syndrome after native valve-sparing aortic valve surgery" and concluded "In summary, blood type 0, valve-sparing root replacement and peak C-reactive protein>15 mg/dl within 48 hours postoperatively are significantly associated with postpericardiotomy syndrome after native valve-sparing aortic valve surgery. Particularly, the combination of all three risk factors is linked to a prohibitive risk of postpericardiotomy syndrome.". I have the following comments:

Response: We would like to thank the reviewer very much for taking the time and effort to review our manuscript and for providing suggestions on how to improve the current draft.

Comment 1: Please don't include number of patients in the methods and move that to abstract results.

Response 1: We would like to thank the reviewer for this remark. 

Changes 1: We have accordingly adapted the Methods and Results sections of the Abstract (see lines 24-35).

Comment 2: Please add a definition with reference to "postpericardiotomy syndrome".

Response 2: We would like to thank the reviewer for this suggestion. As stated in the Introduction, post-pericardiotomy syndrome is an autoinflammatory/autoimmune syndrome with pleuropericardial involvement affecting approximately 10-30% of patients in the early postoperative period (i.e., that is up to three months) after cardiac surgery [1, 2]. At our institution, post-pericardiotomy syndrome was or rather is diagnosed if patients show at least two of the following diagnostic criteria as adapted from Imazio et al. [3]: (I) evidence of new or worsening pericardial effusion on TTE, (II) evidence of new or worsening pleural effusions on lung ultrasound and chest X-ray, (III) pleuritic chest pain, (IV) fever or (V) elevated inflammatory markers without alternative causes. Moreover, nosocomial systemic infection has to be ruled out.

[1] Imazio M, Brucato A, Rovere ME, Gandino A, Cemin R, Ferrua S, et al (2011) Contemporary features, risk factors, and prognosis of the post-pericardiotomy syndrome. Am J Cardiol 108(8):1183-7. https://doi.org/10.1016/j.amjcard.2011.06.025

[2] Imazio M, Brucato A, Ferrazzi P, Pullara A, Adler Y, Barosi A, et al (2014) Colchicine for prevention of postpericardiotomy syndrome and postoperative atrial fibrillation: the COPPS-2 randomized clinical trial. Jama 312(10):1016-23. https://doi.org/10.1001/jama.2014.11026

[3] Imazio M, Brucato A, Ferrazzi P, Spodick DH, Adler Y (2013) Postpericardiotomy syndrome: a proposal for diagnostic criteria. J Cardiovasc Med (Hagerstown) 14(5):351-3. https://doi.org/10.2459/JCM.0b013e328353807d

Changes 2: We have adapted the definition and diagnostic criteria for PPS (see line 54 as well as lines 111-115).

Comment 3: Table 3: please acknowledge the wide 95%CI in your multivariable model.

Response 3: We would like to thank the reviewer for this remark. We agree that the wide 95% CIs deserve some acknowledgement.

Changes 3: We have accordingly adapted the Limitations section (see lines 321-324).

Comment 4: Language revision is essential eg "blood type 0," should be "blood type O".

Response 4: We would like to thank the reviewer for this remark. 

Changes 4: This minor revision has been implemented in the revised version of the manuscript.

Comment 5: In the conclusion, please revisit and edit this sentence "combination of all three risk factors is linked to a prohibitive risk of PPS".

Response 5: We would like to thank the reviewer for this suggestion. As illustrated in Fig 1, 73% of patients displaying all three risk factors developed PPS in our cohort. Hence, patients in whom all three risk factors are present are at a particularly high risk for PPS and should receive preventive anti-inflammatory medical treatment. However, we agree that the term “prohibitive” needs some revision as some readers could assume that surgery should not be performed at all in such patients which is clearly not the case. 

Changes 5: We have accordingly revised the conclusion (see lines 337-338).

Comment 6: Please cite the following relevant randomized trial in your discussion (PMID: 34788640).

Response 6: We would like to thank the reviewer for this suggestion. 

Changes 6: We incorporated the above-mentioned trial into our discussion, citing it within the context of retained blood syndrome (see lines 297-301).

Reviewer 2:

Comment: Theresa Holst and colleagues reported on 91 consecutive patients who underwent native valve-sparing aortic valve surgery and the incidence of post-pericardiotomy syndrome (PPS) among them. I congratulate them on achieving good surgical outcomes and handling PPS well. The authors attempted to identify predictive factors for PPS through the clinical data of these patients. However, there are several issues with the article.

Response: We would like to thank the reviewer very much for these kind remarks.

Comment 1: The study included 91 patients, which is a small sample size that highlighted considerable heterogeneity among the patients. They underwent various types of native valve-sparing aortic valve surgeries, and some patients also had concomitant procedures, such as mitral and tricuspid valve surgeries, and CABG. Therefore, the trauma experienced by these patients was inconsistent and difficult to assess. For example, did patients undergoing VSRR also have additional surgeries, leading to the occurrence of PPS?

Response 1: We would like to thank the reviewer for these important remarks. 

As displayed in Table 1, concomitant procedures were performed in 18/91 (20%) of patients undergoing native valve-sparing aortic valve surgery. 3/18 (17%) patients with concomitant procedures developed PPS postoperatively, while 15/18 (83%) had an uneventful postoperative course with respect to PPS. In comparison, 18/73 (25%) of patients undergoing isolated native valve-sparing aortic valve surgery developed PPS, while 55/73 (75%) did not (p=0.551). 

In total, 35/91 patients underwent valve-sparing root replacement (VSRR). Of the 35 VSRR patients, 12 (34%) underwent concomitant procedures while 23 (66%) underwent isolated VSRR. Only 3/12 (25%) VSRR patients with concomitant procedures developed PPS postoperatively (concomitant procedures: 1x mitral valve repair, 1x tricuspid valve repair + closure of an atrial septal defect and 1x left atrial appendage occlusion), while 9/12 (75%) VSRR patients with concomitant procedures had an uneventful postoperative course with respect to PPS. In comparison 11/23 (48%) of patients undergoing isolated VSRR developed PPS while 12/23 (52%) did not (p=0.282).

Our data, hence, indicate that the risk of developing PPS postoperatively does not appear to be increased if concomitant procedures are performed alongside native valve-sparing aortic valve surgery in general or VSRR in specific. 

Moreover, most previous large studies on PPS after cardiac surgery included patients undergoing a wide range of surgical procedures for distinctly different pathologies [1-4]. Compared to these studies, our study cohort appears quite homogenous as all our 91 patients were non-elderly patients that were primarily referred to us for native valve-sparing aortic valve surgery for severe aortic valve disease. In 18 of these, preoperative diagnostics revealed concomitant pathologies (e.g. at least moderate mitral or tricuspid valve disease, ≥70% coronary artery stenosis, atrial septal defect, patent foramen ovale, persistent or paroxysmal atrial fibrillation) that needed to be addressed simultaneously intraoperatively in accordance with current guidelines. Excluding these 18 patients from our analysis would have significantly reduced our sample size limiting the generalization of our findings even further. Therefore, we decided to include these patients in our analysis. 

[1] Imazio M, Brucato A, Rovere ME, Gandino A, Cemin R, Ferrua S, et al (2011) Contemporary features, risk factors, and prognosis of the post-pericardiotomy syndrome. Am J Cardiol 108(8):1183-7. https://doi.org/10.1016/j.amjcard.2011.06.025

[2] Imazio M, Brucato A, Ferrazzi P, Pullara A, Adler Y, Barosi A, et al (2014) Colchicine for prevention of postpericardiotomy syndrome and postoperative atrial fibrillation: the COPPS-2 randomized clinical trial. Jama 312(10):1016-23. https://doi.org/10.1001/jama.2014.11026

[3] Pan T, Jiang CY, Zhang H, Han XK, Zhang HT, Jiang XY, et al (2023) The low-dose colchicine in patients after non-CABG cardiac surgery: a randomized controlled trial. Crit Care 27(1):49. https://doi.org/10.1186/s13054-023-04341-9

[4] Gabaldo K, Sutlić Ž, Mišković D, Knežević Praveček M, Prvulović Đ, Vujeva B, et al (2019) Postpericardiotomy syndrome incidence, diagnostic and treatment strategies: experience AT two collaborative centers. Acta Clin Croat 58(1):57-62. https://doi.org/10.20471/acc.2019.58.01.08

Changes 1: As our data indicate that the risk of developing PPS postoperatively does not appear to be increased if concomitant procedures are performed alongside native valve-sparing aortic valve surgery in general or VSRR in specific, no further change to the original manuscript has been made with respect to this aspect. However, if desired the above-mentioned information could be added as supplementary materials. To underline the homogeneity of the study cohort in contrast to most previously published reports, the Materials and Methods section has been revised (see lines 88-103).

Comment 2: In Table 1, the statistical methods used are confusing. For categorical tests, especially with small sample sizes, using Fisher's exact test and adopting a two-tailed significance for the P-value would be more appropriate. However, in practice, the P-value for AV morphology seems to have been calculated using a one-tailed significance from Fisher's exact test (P < 0.05), but when corrected for two-tailed significance (P = 0.056), which makes it less significant. Additionally, the P-value for VSRR, despite showing a significant difference, should be 0.004 instead of 0.002. Similar statistical calculations should be re-examined by the authors, as this could lead to entirely inconsistent conclusions.

Response 2: We would like to thank the reviewer for this insightful comment. 

As far as we know, the only two strict requirements that need to be met in order to be able to apply the chi-square test to categorical variables are the following:

(1) Expected frequencies for each cell are at least 1

(2) Expected frequencies should be at least 5 for the majority (80%) of the cells

Those requirements are most likely to be met if the sample size equals at least the number of cells multiplied by 5. However, it is certainly true that larger sample sizes improve accuracy of the results.

If both requirements were met, we used the chi-square test (e.g. for hypertension, known allergies, blood type O, blood type A, AV morphology, surgical access, isolated AV repair, VSRR, use of Dacron, need for blood transfusions). If either one or both requirements were not met, we used Fisher’s exact test adopting a two-tailed significance for the p-value in all cases. A one-tailed significance has not been adopted in any case. 

The complete abandonment of the chi-square test and the application of Fisher’s exact test in all categorical variables would lead to the results displayed in the "Response to Editor and Reviewers Document" (corrected p-values marked by an asterisk).

The only relevant difference to results from the chi-square test would be the fact that the p-value for AV morphology would increase from 0.040 to 0.056. Correctly speaking, this difference would not be considered statistically significant anymore. Yet, a strong tendency towards a relevant difference would certainly remain. Regarding all other variables, p-values yielded by Fisher’s exact test instead of chi-square test would remain either greater than or less than 0.05 and would not change significantly. Moreover, AV morphology, more precisely tricuspid AV morphology, was also identified as significantly associated with PPS by univariate binary logistic regression. Hence, the subsequent steps of the statistical analysis and conclusions drawn would still remain consistent and would not have to be changed if we adopted Fisher’s exact test for all categorical variables. 

Changes 2: We have reexamined statistical calculations adopting Fisher’s exact test for all categorical variables and replaced p-values formerly yielded by chi-square test by p-values derived from Fisher’s exact test (see e.g. Tables 1 and 2).

Comment 3: Regard as the results, the authors identified some predictors for PPS, such as blood type O, VSRR, and peak CPR > 15 mg/dl within 48 hours. Although these results showed statistical differences, I have some reservations about these factors. Firstly, the authors did not explain in detail in the discussion why blood type O is a predictive factor; moreover, this result differs from previous studies (blood type B), and we are very curious about why different results emerged, or is it just a statistical coincidence? Secondly, the article explains that VSRR is a predictor for PPS but cannot be explained by surgical trauma, as ROSS surgery, which involves a similar level of trauma, did not show positive results. So, what are the specific mechanisms of VSRR leading to PPS? Besides, how is peak CPR defined? Is it limited to a continuous rise within 48 hours to reach a peak, or does the peak CPR during the entire hospital stay occur within 48 hours post-surgery? If it's the latter, this indicator's reference value for clinical practice is significantly reduced, as you never know if the next blood draw will be the peak, and PPS can already be detected using echocardiography before detecting the peak.

Response 3: We would like to thank the reviewer for this kind review and an important comment. 

There is evidence that patients with certain blood types may be predisposed to infections and autoimmune conditions. Yet, results of studies examining the link between ABO blood group and autoimmunity have been inconsistent, often due to limited samples sizes [1-4]. To our knowledge, only two previous studies reported a potential association of ABO blood group with post-pericardiotomy syndrome: Maisch et al. found anti-endothelial antibodies to be more frequent in patients who developed PPS than in patients who did not. Moreover, they suggested an association of the prevalence of this specific form of autoantibodies after cardiac surgery with blood groups A, B and O, but not with AB [5]. Miller et al. reported p

---

## [Editor Report · Decision Letter 1]

17 Jun 2024

Predictors of post-pericardiotomy syndrome after native valve-sparing aortic valve surgery

PONE-D-24-05122R1

Dear Dr. Holst,

We’re pleased to inform you that your manuscript has been judged scientifically suitable for publication and will be formally accepted for publication once it meets all outstanding technical requirements.

Kind regards,

Eyüp Serhat Çalık

Academic Editor

PLOS ONE
---

## [Editor Report · Acceptance letter]

19 Jun 2024

PONE-D-24-05122R1 

PLOS ONE

Dear Dr. Holst, 

I'm pleased to inform you that your manuscript has been deemed suitable for publication in PLOS ONE. Congratulations! Your manuscript is now being handed over to our production team.

Kind regards, 

on behalf of

Dr. Eyüp Serhat Çalık 

Academic Editor

PLOS ONE